# Reduced Graphene Oxide Modified Nitrogen-Doped Chitosan Carbon Fiber with Excellent Electromagnetic Wave Absorbing Performance

**DOI:** 10.3390/nano14070587

**Published:** 2024-03-27

**Authors:** Mengyao Guo, Ming Lin, Jingwei Xu, Yongjiao Pan, Chen Ma, Guohua Chen

**Affiliations:** College of Materials Science and Engineering, Huaqiao University, Xiamen 361021, China; 1614111009@stu.hqu.edu.cn (M.G.); 21014081077@stu.hqu.edu.cn (M.L.); 23011081017@stu.hqu.edu.cn (J.X.); 2014141019@stu.hqu.edu.cn (Y.P.)

**Keywords:** chitosan, electrospinning, graphene, carbon nanofiber, electromagnetic wave absorption

## Abstract

Lightweight and low-cost one-dimensional carbon materials, especially biomass carbon fibers with multiple porous structures, have received wide attention in the field of electromagnetic wave absorption. In this paper, graphene-coated N-doped porous carbon nanofibers (PCNF) with excellent wave absorption properties were successfully synthesized via electrostatic spinning, electrostatic self-assembly, and high-temperature carbonization. The obtained results showed that the minimum reflection loss of the absorbing carbon fiber obtained under the carbonization condition of 800 °C is −51.047 dB, and the absorption bandwidth of reflection loss below −20 dB is 10.16 GHz. This work shows that carbonization temperature and filler content have a certain effect on the wave-absorbing properties of fiber, graphene with nanofiber, and the design and preparation of high-performance absorbing materials by combining the characteristics of graphene and nanofibers and multi-component coupling to provide new ideas for the research of absorbing materials.

## 1. Introduction

With the rapid development of communication technology, electronic devices bring us much convenience while also generating electromagnetic waves in various frequency bands in our living environment [1,2,3]. Electromagnetic wave pollution not only threatens human physiological health but also interferes with the normal operation of sensitive electronic equipment [4]. Therefore, it is important to eliminate electromagnetic interference to ensure the safety of operators and the normal operation of sensitive systems [5]. Various types of conductive materials (including carbon nanotubes [6], graphene [7], carbon fibers [8], MXene [9], etc.) and magnetic materials (Fe [10], Co [11], Ni [9,12], ferrite [13], alloys [14], etc.) have been used to dissipate electromagnetic waves based on their own dielectric or magnetic losses. Magnetically responsive materials have better electromagnetic wave absorption capability at a wider frequency band [15]. However, magnetic absorbing materials also have the disadvantages of high density [16], poor corrosion resistance [17], and low Curie temperature [18]. Dielectric fillers can make up for these deficiencies. Dielectric loss materials, such as carbon materials, MXene, SiC, and conductive polymers, have good electronic conductivity. They are lightweight, low-density, corrosion resistant, high conductivity, and good thermal stability. Therefore, dielectric loss materials have been known as excellent materials for absorbing and shielding electromagnetic waves [12,19].

Among these carbon materials, graphene, especially reduced graphene oxide (rGO), has emerged in the field of electromagnetic wave absorption with its large specific surface area, structural defects, and low density [20,21,22]. The large specific surface area of rGO effectively reduces the concentration in the overall conductive network for fabricating absorbing materials. The polarization centers generated by defects and surface groups cause more dipole relaxation losses [23]. Most researchers usually adopt the compounding of rGO with some magnetic materials to improve their impedance matching and enhance the absorbing performance of the materials. For instance, the reduced graphene–nickel nanocomposites prepared by Zhang et al. [24] using gamma-ray irradiation had a minimum reflection loss of −24.8 dB and a maximum effective absorption bandwidth of 6.9 GHz with a layer thickness of 9 mm. Ge et al. [25] prepared accordion-like graphene with iron nanoparticles distributed between layers by an in situ growth method, and the minimum reflection loss reached −54.6 dB. However, the disadvantages of magnetic absorbing materials, such as high density, high thickness, and low Curie temperature, inhibit their practical applications. In addition, structural size regulation of wave-absorbing materials is also an effective method to improve the wave absorption performance of materials [26]. Carbon nanofibers show a large potential for application in the field of electromagnetic wave absorption due to their low density and high aspect ratio [27]. While ensuring the advantages of carbon materials, biomass carbon nanofibers also have wide availability and economic feasibility, which is suitable for the development of electromagnetic wave-absorbing materials. Du et al. [28] prepared lignin-based carbon nanofibers by electrospinning and carbonization process; the minimum reflection loss value can reach −41.4 dB. Pei et al. [29] dispersed carbon nanotubes (CNTs) evenly in a chitosan matrix, and the CNTs/CS 3D printing ink showed an excellent electromagnetic shielding effect.

To solve the problems of high density, high thickness, and low Curie temperature of current magnetic nano-absorbing materials, biomass carbon materials and reduced graphene oxide were combined into core-shell nanofibers. In this work, chitosan (CS) was used as raw material to prepare nanofibers by electrostatic spinning. GO was encapsulated on the fiber surface using the electrostatic self-assembly principle impregnation process, and the precursor fibers were obtained after reduction. Porous carbon nanofibers (PCNF) with excellent wave absorption properties were obtained after high-temperature carbonization, which integrates the characteristics of graphene and nanofibers as well as a multi-component coupling. The design and preparation of high-performance wave-absorbing materials were carried out by combining graphene and nanofibers and multi-component coupling. The carbon nanofibers with the outer rGO layer and inner porous layer were obtained by the carbonization process. Then, the morphology, structure, and substance of the material were analyzed; also, the contrast experiments of different carbonization temperatures and different mass ratios were set up. The electromagnetic parameters of the experimental group were tested by a vector network analyzer, and the optimal absorbing performance parameters were obtained. Then, the wave-absorbing mechanism was analyzed using the electromagnetic parameters.

## 2. Experimental

### 2.1. Materials

Chitosan (CS, average Mw ≈ 300,000) bulks were purchased from Zhejiang Jinke Pharmaceutical Co., Ltd., Shangyu, China. Poly (ethylene oxide) (PEO, average Mw ≈ 1,000,000) powders and paraffin wax (pathological grade, melting point 58~60 °C) were provided by Aladdin Industrial Co., Ltd., Shanghai, China. Acetic acid (HAc, content ≥ 99.0%) and ethanol (mass fraction ≥ 99.0%) were obtained from Sinopharm Chemical Reagent Co., Ltd., Shanghai, China. All the chemical solvents were of analytical grade and used without further purification.

### 2.2. Preparation of GO Dispersion

GO was prepared by the modified Hummers method [30] and dispersed in 90% ethanol solution with 0.01 wt% at room temperature. The dispersive solution of GO was prepared by ultrasonification with a high-frequency multifunctional apparatus for 15 min to ensure the uniformity of GO dispersion.

### 2.3. Preparation of Electrospinning Solutions

An aqueous solution of acetic acid with 90% concentration was prepared. According to the ratio of CS/PEO as 9/1, the powder of CS and PEO were weighed and evenly mixed in 90% acetic acid aqueous solution to form a spinning solution with a mass fraction of 9 wt% to ensure uniform mixing. The solution was stirred in a magnetic agitator for 9 h at room temperature. Let the solution stand for 10 min to remove the bubble in the spinning solution. The obtained solution is ready for subsequent electrostatic spinning. In addition, the principle of mixing and using is adopted, which can be used up in 5 h after preparation to prevent the layering of the spinning solution and ensure the uniformity of the solute to obtain a uniform chitosan protofilament.

### 2.4. Preparation of Fibers

Use the electrostatic spinning machine, cover the aluminum foil paper on the receiver of the electrostatic spinning machine, use a 5 mL injector to absorb the spinning liquid, use a 21# flat metal needle on the syringe head, put the syringe on the injection device, and clip the high voltage power supply on the metal needle. Electrospinning was carried out by adjusting the receiving distance to 15 cm, the injection speed to 0.2 mm/min, and the voltage to 20 kV. After spinning, the fiber felt was separated from the aluminum foil, dried in the oven at 60 °C, and kept in the dryer.

The CS/PEO fiber felt was gently placed into the ethanol-dispersing solution with tweezers. The dispersing solution was absorbed with the glue head dropper and slowly added to the surface of the fiber felt to make it fully infiltrated. After dipping for 1 min, the dispersing liquid was sucked out with an eyedropper. The fiber felt was frozen in the refrigerator and then freeze-dried in a cold dryer for 12 h to obtain GO-coated CS/PEO/GO nanofibers.

The CS/PEO/GO fiber felt was put into the UV oven (Power 1000 W, FXLite 600F, FUWO, Guangzhou, China) for UV light reduction. The reduction time was 4 h. The sample was turned over every 2 h, and the GO coated on the fiber surface was reduced to rGO to obtain CS/PEO/rGO nanofibers.

The reduced CS/PEO/rGO fibers were put into the tubular furnace and carbonized at high temperatures under an N_2_ atmosphere. Before heating up, inject N_2_ for 15 min to exhaust the air in the tube furnace. In addition, only a single sample is put into each carbonization to ensure uniform heating. The initial temperature was set at 30 °C, the heating rate was 5 °C/min, and the final temperature was 800 °C, 900 °C, and 1000 °C, respectively. The carbonization time was 60 min. The porous carbon fibers PCNF-800, PCNF-900, and PCNF-1000 were prepared. (See Figure 1).

### 2.5. Characterization

The morphology and microstructure of fibers were observed by a scanning electron microscope (SEM, Hitachi S-4800, Rock Hill, SC, USA) at an accelerating voltage of 10 kV and a transmission electron microscope (TEM, JEOL JEM-2100, Grand Rapids, MI, USA) at an accelerating voltage of 200 kV. The structure and defect degree of the samples were identified by the Raman spectrometer (Raman, HJY LABRAM, Essex, UK). X-ray photoelectron spectroscopy (XPS, TMO Thermo, ESCALAB 250XI, Waltham, MA, USA) analyzed the composition of elements inside the sample. The absorbing properties of the samples were tested by a vector network analyzer (VNA, Keysight Agilent E5071C, Santa Rosa, CA, USA). The electromagnetic parameters and absorption performance test details are shown in Appendix A in Appendix A. 

## 3. Results and Discussion

### 3.1. Macro-Morphology of Film

The PCNF-900 film was cut into a rectangular shape of 3 cm and 1 cm wide. Figure 2 shows the optical picture of PCNF-900. It can be seen from the figure that the carbon fiber membrane obtained by carbonization at 900 °C has a certain flexibility and bending strength. After folding and bending once, the carbon fiber membrane will not crack and can be restored to its original state after bending 10 times. There is no obvious crease in the bending, indicating that the carbon fiber membrane has good mechanical strength, which can expand the advantage of the material.

### 3.2. Structural Characterization

The micro-morphologies of CS/PEO/GO and CS/PEO/rGO fibers are shown in Figure 3a,b, and the arrows in the figure indicate the location of graphene. After impregnation with GO dispersion, there are still a lot of pores between the fibers. The CS/PEO/GO fiber surface is evenly coated with thin GO film layers, and the fiber surface is relatively smooth. In contrast, there is little difference in the surface morphology of fiber dimension before and after reduction; the fiber surface after reduction is rougher than before, which is due to the different properties of GO and rGO; GO is more flexible than rGO.

Figure 3c–f shows the SEM of PCNF-800, PCNF-900, and PCNF-1000. It can be seen from the figure that CS/PEO/rGO is carbonized at 800 °C, and the rGO on the fiber surface cracks and partially falls off. The internal structure of the fiber is relatively tight, so it is impossible to observe whether there is a porous structure inside the fiber. Figure 3d,e is SEM images of PCNF-900, similar to Figure 3a. After high-temperature carbonization, rGO cracking occurs on the fiber surface, and rGO has fallen off in some areas, but more rGO is still attached to the fiber surface. In section diagram (c), it can be directly observed that the fiber inside PCNF-900 has an obvious porous structure. Figure 3d is the SEM image of PCNF-1000. It can be found that the rGO attached to the fiber surface peels off significantly, and a large number of pores appear in the fiber interior. During the carbonization step, the chitosan fiber would decompose and form a porous structure. This process could result in the volume expansion of fiber, which is the main reason causing the peeling in rGO. With the increase in carbonization temperature, the decomposition and expansion of chitosan fiber also become severe. Therefore, rGO can hardly be found on the fiber surface at 1000 °C.

According to TEM test observation, as shown in Figure 3g,h, the surface of the fiber is coated with several GO layers. Because GO laminates are thin and transparent and CS/PEO cores are thick, there is a large contrast between them. It can be seen from the figure that the impregnated fibers form a core-shell fiber structure coated by GO laminates.

Figure 3i–k shows the TEM images of PCNF-800, PCNF-900, and PCNF-1000. It can be seen from the figure that compared with PCNF-800, PCNF-900 and PCNF-1000 have more loose fibers inside, and PCNF-800 is still coated with a large amount of rGO. The amount of rGO on the surface of PCNF-900 and PCNF-1000 decreased in turn, which was consistent with the SEM test results of the upper part, indicating that in the process of high-temperature carbonization, the rGO attached to the fiber surface accelerated shedding with the increase of carbonization temperature, and the pore structure in the fiber also increased with the increase of carbonization temperature.

### 3.3. Component Characterization

To prove that the carbonized chitosan fiber still retains a high proportion of N element, the chemical elements and properties of PCNF-900 surface were determined by X-ray spectrometer. As shown in Figure 4a, a characteristic peak of N (~400 eV) appears obviously on the XPS spectrum of PCNF-900. As shown in Figure 4b, the N 1s XPS spectrum can be divided into two peaks, which are 398.33 eV and 401.03 eV, respectively. The peak at 398.33 eV can be attributed to neutral imine nitrogen atoms produced in the polymer network [31]. The peak at 401.03 eV belongs to positively charged N, and the presence of this positively charged nitrogen can be explained by the fact that the polymer can be altered by an X-ray beam to produce a positive charge through electron ionization [32]. The atomic ratio was calculated by peak fitting, and the XPS atomic ratio of PCNF-900 was sorted out, as shown in Table 1. The XPS spectral analysis of PCNF-900 shows that the ratio of N atoms can reach 2.67%, indicating that chitosan, as the only nitrogen source, can provide sufficient N doping for carbon fiber, which increases the internal defects of carbon fiber. The polarization center generated by these defects causes dipole relaxation loss, which greatly increases the absorption loss ability of carbon fiber to electromagnetic waves.

To test the degree of defect in the material, Figure 5 shows the Raman spectra of CS/PEO-900 and PCNF-900. It can be seen from the figure that both of them have unique peaks, D and G, of carbon materials. Origin software is utilized to use the Gaussian equation to fit the two peaks. Then, the integration was used to obtain the peak area of peak D and peak G of the two samples, and the R-value of the two samples was calculated by using R = *I_D_*/*I_G_*. The R-value of CS/PEO-900 was 1.0730, and that of PCNF-900 was 1.3942. In addition, the R-value of PCNF-900 is much higher than that of CS/PEO-900, indicating that the addition of rGO increases the defects of carbon fiber. It is speculated that these holes can provide sites for the consumption of electromagnetic waves.

Figure 6 shows the Raman spectra of PCNF-800, PCNF-900, and PCNF-1000. It is found that under different carbonization temperatures, the R-value of carbon fiber changes. The R-value of PCNF-800 is 1.3584, and that of PCNF-900 is 1.3942. The R-value of PCNF-1000 is 1.5569, as shown in Figure 6d. The R-value of carbon fiber increases with the increase of carbonization temperature, indicating that with the increase of carbonization temperature, the disorder degree of carbon fiber becomes larger, and the internal defects of the fiber increase, which can provide conditions for the consumption of electromagnetic waves.

### 3.4. EMW Absorption Capacity

Figure 7 shows the relationship between reflection loss value and incident frequency of PCNF-800 with different mass ratios. PCNF-800 samples were mixed with paraffin according to the mass ratio of 1/19 (that is, the mass fraction of carbon fiber samples was 5%), 1/9 (10%), and 3/7 (30%), respectively. It was put into the vector network analyzer for testing. The test results showed that under the carbonation temperature of 800 °C, the carbon fiber sample had the best wave absorption performance under the mass fraction of 10%. As shown in Figure 7b, the PCNF-800 (10%) sample has a minimum reflection loss of −51.047 dB at 9.28 GHz with a thickness of 2.9 mm, and an absorption bandwidth of 10.16 GHz with a reflection loss below −20 dB, the frequency range is 4.32–14.48 GHz. (a) The PCNF-800 (5%) sample in the figure has a minimum reflection loss of −15.475 dB at 6.4 GHz at a thickness of 4.9 mm. (c) The PCNF-800 (30%) sample in the figure has a minimum reflection loss of −5.416 dB at 18 GHz when the thickness is 0.9 mm.

Figure 8 shows the relationship between reflection loss value and incident frequency of PCNF-900 with different mass ratios. Samples of PCNF-900 were mixed with paraffin wax according to the mass ratio of 1/19 (5%), 1/9 (10%), and 3/7 (30%), respectively, and pressed into an axial structure with molds. It was put into the vector network analyzer for testing. The test results showed that the carbon fiber sample had the best absorbing performance at the carbonization temperature of 900 °C and the mass fraction of 10%. As shown in Figure 8b, the sample of PCNF-900 (10%) has a minimum reflection loss of −47.914 dB at 4.08 GHz when the thickness is 5.4 mm and also has a smaller reflection loss of −43.310 GHz when the thickness is 2 mm. The absorption bandwidth of the sample with reflection loss below −20 dB is 12.08 GHz, and the frequency range is 3.84 to 15.92 GHz. (a) The PCNF-900 (5%) sample in the figure has a minimum reflection loss of −17.439 dB at 17.76 GHz at a thickness of 5.5 mm. The PCNF-900 (30%) sample in Figure 8c has a minimum reflection loss of −9.839 dB at 16.32 GHz at 1 mm thickness. In conclusion, compared with some other carbon-based nano-absorbing materials, PCNF-900 not only has low loading capacity and thin matching thickness but also has stronger microwave absorption energy in a wider frequency range. Therefore, it is expected to be used as a lightweight, thin-layer microwave absorbing material in the middle and high-frequency range.

Figure 9 shows the relationship between reflection loss value and incident frequency of PCNF-1000 with different mass ratios. Samples of PCNF-1000 were mixed with paraffin wax at the mass ratio of 1/19 (5%), 1/9 (10%), and 3/7 (30%), respectively, and pressed into ring structure by mold. It was put into the vector network analyzer for testing. The test results showed that the carbon fiber sample had the best absorbing performance at the carbonization temperature of 1000 °C and the mass fraction of 10%. As shown in Figure 9b, the sample of PCNF-1000 (10%) has a minimum reflection loss of −48.114 dB at 16.8 GHz when the thickness is 1.7 mm, and the absorption bandwidth of the sample whose reflection loss is lower than −20 dB is 3.04 GHz. The frequency ranges are 12.00–12.48 GHz and 15.28–17.84 GHz. (a) The PCNF-1000 (5%) sample in the figure has a minimum reflection loss of −10.792 dB at 18 GHz at a thickness of 5.5 mm. The PCNF-1000 (30%) sample in Figure 9c has a minimum reflection loss of −7.161 dB at 14.56 GHz at 1 mm thickness. Based on the above analysis, it can be seen that although the minimum reflection loss of PCNF-1000 carbon fiber is little different from that of PCNF-800 and PCNF-900, its frequency range below −20 dB is greatly narrowed, so the absorption performance of PCNF-1000 is relatively poor compared with the above two samples. To sum up, the wave-absorbing property of the sample is related to the carbonization temperature, and the wave-absorbing property of the sample decreases when the sintering temperature increases to 1000 °C. Combined with the previous electron microscopy analysis, it can be found that the fiber still maintains a porous structure after carbonization at 1000 °C, but it is difficult to find the distribution of rGO on the surface. It is speculated that the attenuation of rGO absorption performance is related to the shedding of rGO on the fiber surface. In addition, the above three carbonization temperatures at 10% filler content all show excellent electromagnetic wave absorption capacity. It is speculated that the key to the absorption performance of the material is the ability of electromagnetic waves to penetrate the material and the loss of the material to electromagnetic waves. The filler content needs to be in a suitable range. If the filler content is too low, the electromagnetic wave entering the sample cannot be fully absorbed by enough sample loss. On the contrary, too high filler content could result in poor impedance matching, which also affects the absorption performance of the material.

Figure 7, Figure 8 and Figure 9 show that when the thickness changes from 1.0 mm to 5.5 mm, the RL peak moves to the low frequency, which is consistent with the inverse ratio between wavelength and thickness in the one-quarter wavelength theoretical formula:(1)$$t_{m}=nc/(4f_{m}{\left(\left|\varepsilon_{r}\right|\left|\mu_{r}\right|\right)}^{1/2})\;(n=1,\;3,\;5\ldots )$$

Here, *n* is odd, *f_m_* is the resonance frequency, and *t_m_* is the sample thickness. It can be seen that the thickness of the sample is inversely proportional to the resonance frequency. Therefore, the resonance frequency corresponding to the minimum reflection loss peak will continuously move to the low-frequency region with the increase in thickness. Therefore, shifted peaks appear on the graphs for different sample thickness values. The microwave absorption performance can be adjusted by adjusting the sample thickness and the amount of packing of carbon fiber.

Figure 10 shows the relative complex permittivity, complex permeability, and loss angle tangent versus incident frequency for carbon fiber 10 wt% absorbing samples at different carbonization temperatures. The real part of the dielectric constant is an expression of the polarizability of the material, which is mainly caused by dipole polarization and interfacial polarization at microwave frequencies. The microstructure of the outer rGO inner porous carbon nanofiber sample increases the number of dipoles and interfaces, which makes the real part of the dielectric constant of the sample maintain higher values, as shown in Figure 3e. The real part of the dielectric constant fluctuates around 10 for all three. In the PCNF-800 and PCNF-900 samples, the virtual permittivity decreases in the frequency range of 2–18 GHz, and the value is greater than 2, indicating that the amorphous carbon in both samples has some inherent dielectric loss. Therefore, the large dielectric loss of this sample is mainly attributed to the interfacial polarization relaxation loss and ohmic loss caused by the porous carbon nanofiber structure in the outer rGO layer and the inherent dielectric loss of the amorphous carbon. In addition, the imaginary part of the PCNF-1000 sample has a small increasing trend in the range of 14–18 GHz, which is due to the porous fibers containing a large number of defects inside the sample, which generate a large number of carriers and free electrons in the alternating electric field and thus exhibit a strong polarization relaxation. Overall, it seems that the complex permittivity of the sample decreases with the increase in the carbonization temperature. It is known that interfacial polarization, electric dipole polarization, electron polarization, and dielectric relaxation are the main factors affecting the dielectric constant, in addition to the structure, morphology, and size of the crystal, which also affects its value [33,34]. It is presumed that as the carbonization temperature increases, the overall density of carbon fibers increases and the occupied volume in the absorbing sample decreases, making the resistance of the absorbing sample increase and the electrical conductivity decrease according to the free electron theory [35]:(2)$$\varepsilon^{\prime \prime }\approx 1/2\pi \varepsilon_{0}\rho f$$
where *ρ* is the resistivity, *ε*_0_ is the vacuum dielectric constant, and *f* is the frequency of the incident electromagnetic wave, it can be seen that the complex dielectric constant decreases as the resistivity of the material increases. In addition, as the carbonization temperature increases, the various defects inside the fiber decrease, the polarization center in the electromagnetic field also decreases, and the polarization effect inside the material weakens, which also leads to a decrease in the complex permittivity.

In Figure 10b, it can be found that the overall fluctuation range of the real and imaginary parts of the complex permeability of the absorbing samples at different carbonization temperatures is small, basically fluctuating above and below 1 and 0, respectively, which indicates that the magnetic loss of the experimental material in the absorbing process is small, and the phenomenon is basically following the experimental law because no magnetic material is added to the samples, which are mainly carbon fibers.

Figure 10c shows the variation of the dielectric loss tangent and magnetic loss tangent with frequency for the absorbing samples at different carbonization temperatures, which shows that the dielectric loss tangent and magnetic loss tangent both decrease with the increase of carbonization temperature. This indicates that the loss mechanism of the absorbing material is mainly dielectric loss, a phenomenon also seen in other studies [36,37].

In addition, in general, the porous structure can reduce the dielectric constant of the material. It is known that incident waves are insensitive to particles or structures smaller than the sensing wavelength, so porous materials can be considered effective media for mixtures of air and material components. The effective dielectric constant (*ε_eff_*) of porous materials can be determined by the Maxwell–Garnett doctrine [38]:(3)$$\varepsilon_{eff}^{MG}=\varepsilon_{1}\frac{\left(\varepsilon_{2}+2\varepsilon_{1})+2f(\varepsilon_{2}-\varepsilon_{1}\right)}{\left(\varepsilon_{2}+2\varepsilon_{1})-f(\varepsilon_{2}-\varepsilon_{1}\right)}$$
where *ε*_1_ and *ε*_2_ are the dielectric constants of the primary object, respectively, and *f* is the volume fraction of the object in the effective medium. Therefore, the porous structure significantly reduces the effective dielectric constant and facilitates impedance matching.

Figure 11 shows the fiber wave absorption schematic diagram. Chitosan carbonization can generate porous carbon with multi-level pore size distribution. The porous structure can reduce the material density, optimize the impedance matching of the material, and effectively improve the wave absorption performance by inducing interface polarization and inhibiting the eddy current effect. Electromagnetic waves generate multiple reflections and scattering in the porous structure, which is converted into heat consumption. A large number of residual functional groups contained in chitosan carbon can provide polarization sites for electromagnetic wave loss and further improve the absorption efficiency of electromagnetic waves. The chitosan fiber prepared by electrospinning has the advantages of high specific surface area and large porosity, which can better optimize the impedance matching of the material, and electromagnetic waves can easily enter the inside of the material.

The reasons for the excellent wave-absorbing properties of the porous composite carbon fiber are summarized. Firstly, the prepared carbon fiber has a porous structure and a long transmission path, which can further improve the attenuation loss ability of the incident wave. This unique porous structure allows the relative dielectric constant to be adjusted to optimize impedance matching. The defects and interfaces in the porous structure can form multi-polarization, which is also conducive to the dissipation of electromagnetic energy. Secondly, there are a lot of heterogeneous interfaces between carbonized chitosan fiber, rGO, and paraffin wax, which can reduce the power of the incident wave by adjusting the polar bond or charge under the alternating electromagnetic field. Thirdly, the residual defects and oxygen-containing groups on the surface or edge of the rGO as the center of polarization can introduce dipole polarization, which further generates dipole polarization or attenuates the incident wave. Finally, a large number of nitrogen atoms are doped in carbon fiber. Due to the electronegativity difference between carbon atoms and nitrogen atoms, C-N electric dipoles are formed, and dipole polarization losses are generated under alternating electromagnetic fields, further attenuating incident waves. In addition, doped nitrogen atoms acting as donor dopants can provide additional electrons, thus increasing conductive losses.

In summary, this paper uses a wide range of biomass chitosan as the raw material, which has certain advantages in terms of production cost and environmental protection. The synthesis process substitutes the pre-oxidation step of the original fiber carbonization, which has low energy consumption and is free of pollution. Therefore, this study has the potential for wide application.

## 4. Conclusions

In this work, rGO was compounded with chitosan nanofibers by electrostatic spinning and electrostatic self-assembly, and high-temperature carbonization was used to obtain high-efficiency porous carbon nanofiber absorbers. The fiber has a core-shell structure, the outer layer is covered by reduced graphene oxide, and the inner layer is multi-level porous chitosan fiber, which is very conducive to electromagnetic wave absorption. PCNF-900 has a minimum reflection loss of −47.914 dB at 4.08 GHz at 10 wt% filler content, a minimum reflection loss of −43.310 GHz at 2 mm thickness, and an absorption bandwidth of 12.08 GHz at a reflection loss of less than −20 dB. Compared with some other carbon-based nano-absorbing materials, PCNF-900 at 10 wt% filler content not only has low loading and thin matching thickness but also has stronger microwave absorption capability in a wider frequency range and is expected to be put into application as a lightweight, thin-layer microwave absorbing material.

## Figures and Tables

**Figure 1 nanomaterials-14-00587-f001:**
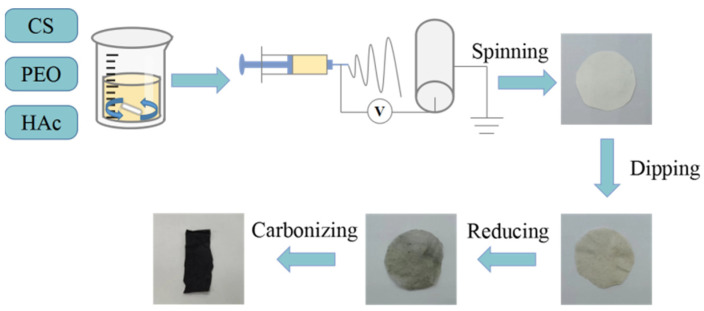
Schematic diagram of the preparation process.

**Figure 2 nanomaterials-14-00587-f002:**
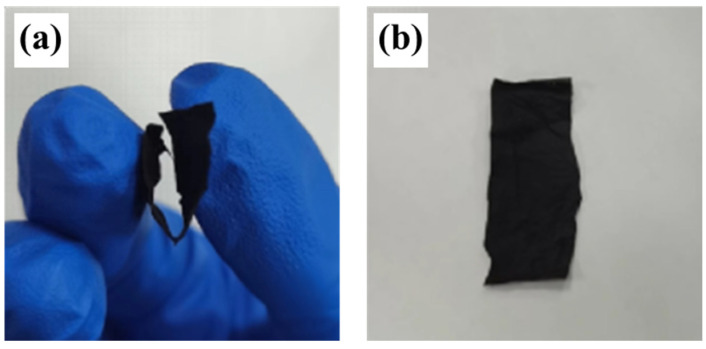
Optical images of the PCNF-900. (**a**) Bending process of the PCNF-900; (**b**) after bending of PCNF-900.

**Figure 3 nanomaterials-14-00587-f003:**
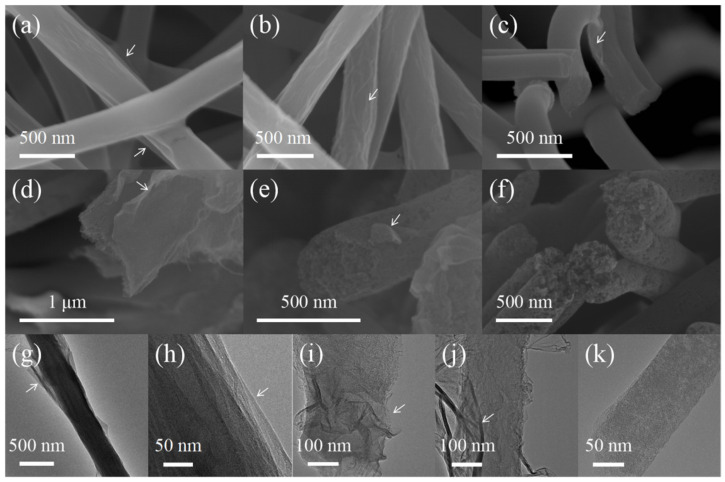
(**a**) SEM of CS/PEO/GO, (**b**) SEM of CS/PEO/rGO, (**c**) SEM of PCNF-800, (**d**,**e**) SEM of PCNF-900, (**f**) SEM of PCNF-1000, (**g**,**h**) TEM of CS/PEO/GO, (**i**) TEM of PCNF-800, (**j**) PCNF-900, (**k**) PCNF-1000.

**Figure 4 nanomaterials-14-00587-f004:**
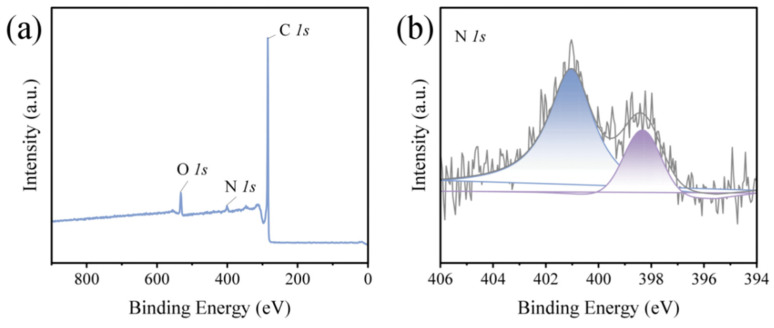
(**a**) XPS broad scan spectra of PCNF-900, (**b**) XPS N 1s curves of PCNF-900.

**Figure 5 nanomaterials-14-00587-f005:**
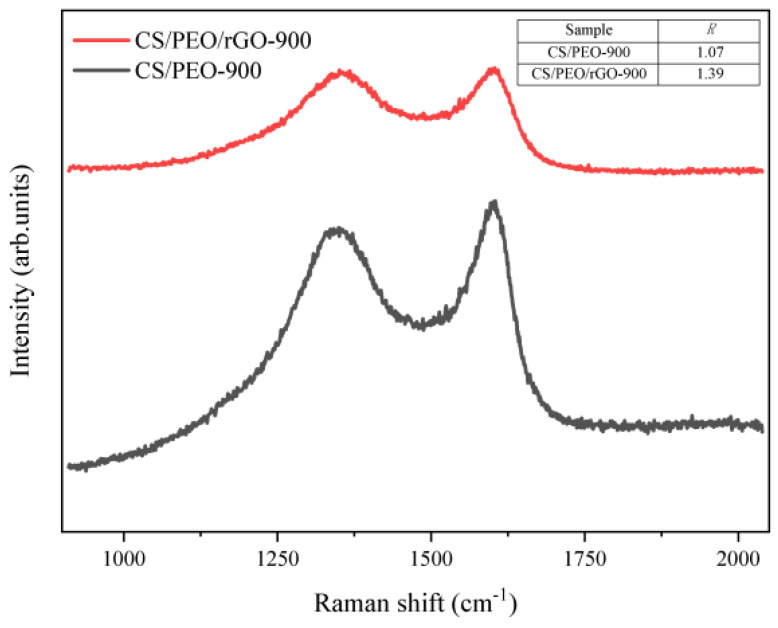
Raman spectra of CS/PEO-900 and PCNF-900.

**Figure 6 nanomaterials-14-00587-f006:**
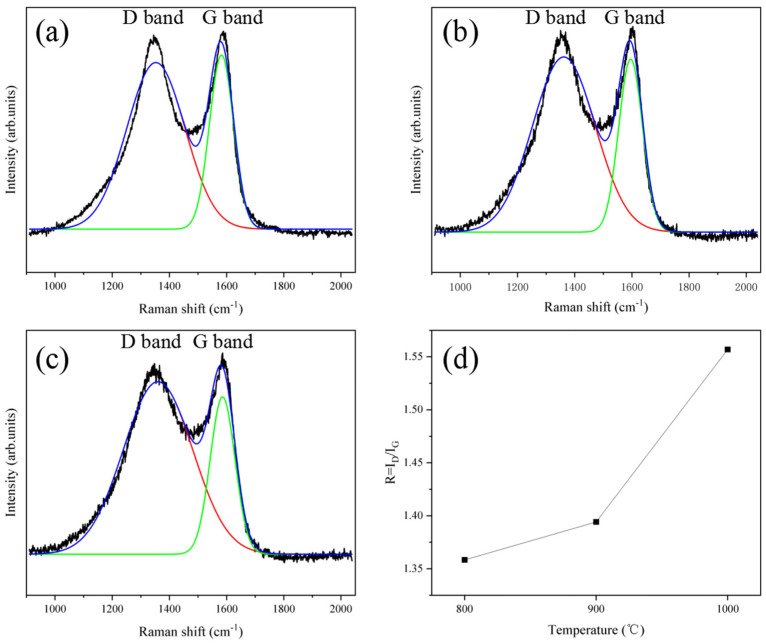
Raman spectra of PCNF-800 (**a**), PCNF-900 (**b**), and PCNF-1000 (**c**); (**d**) R-value trend chart of PCNF-800, PCNF-900, and PCNF-1000.

**Figure 7 nanomaterials-14-00587-f007:**
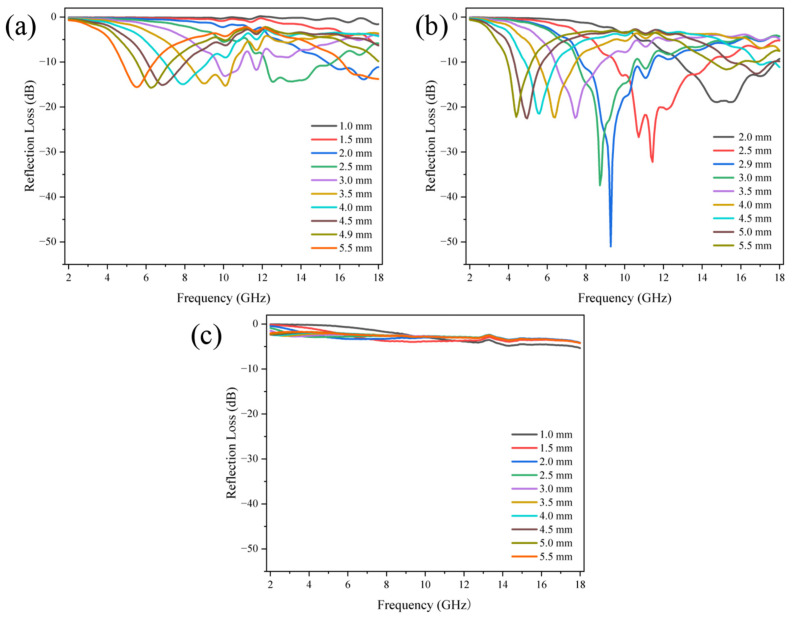
The relationship between reflection loss and incident frequency of PCNF-800 with different mass ratios. (**a**) 5 wt%; (**b**) 10 wt%; (**c**) 30 wt%.

**Figure 8 nanomaterials-14-00587-f008:**
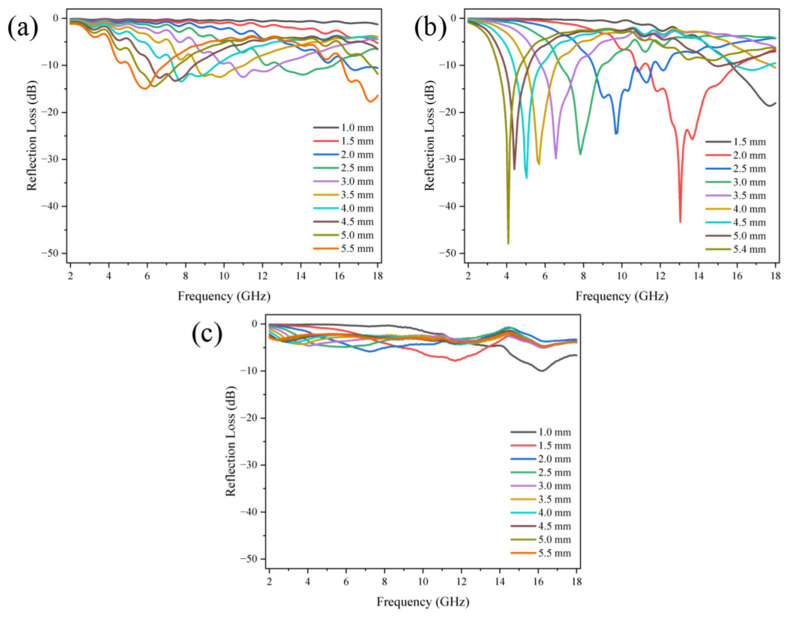
The relationship between reflection loss and incident frequency of PCNF-900 with different mass ratios. (**a**) 5 wt%; (**b**) 10 wt%; (**c**) 30 wt%.

**Figure 9 nanomaterials-14-00587-f009:**
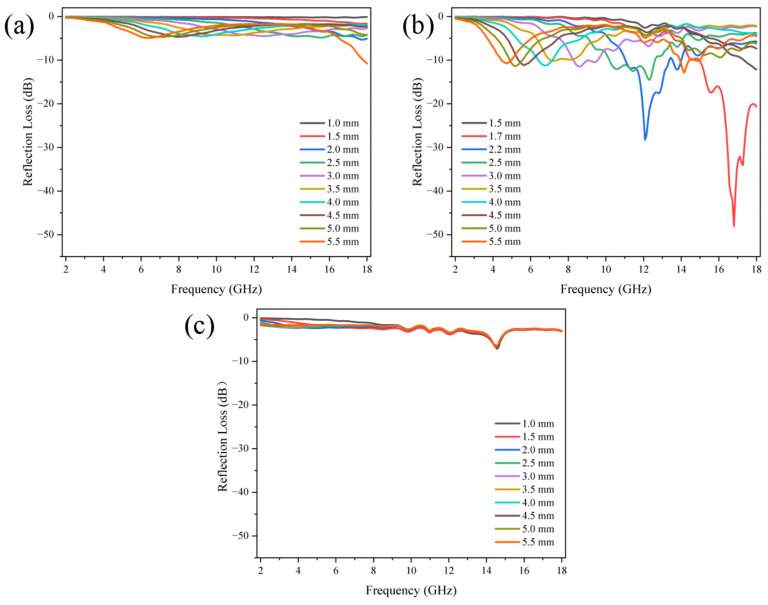
The relationship between reflection loss and incident frequency of PCNF-1000 with different mass ratios. (**a**) 5 wt%; (**b**) 10 wt%; (**c**) 30 wt%.

**Figure 10 nanomaterials-14-00587-f010:**
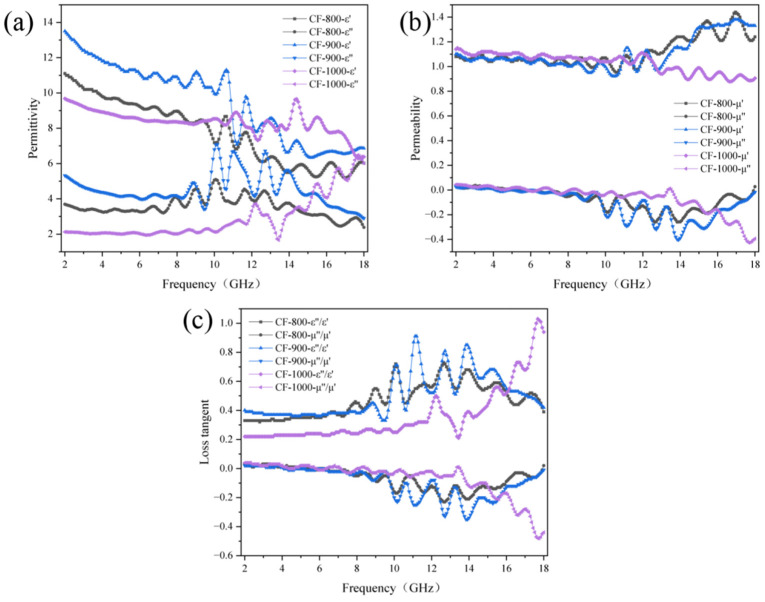
The relationship between electromagnetic parameters and incident frequency of carbon fiber. (**a**) Relative complex dielectric constant; (**b**) complex permeability; (**c**) loss angle tangent.

**Figure 11 nanomaterials-14-00587-f011:**
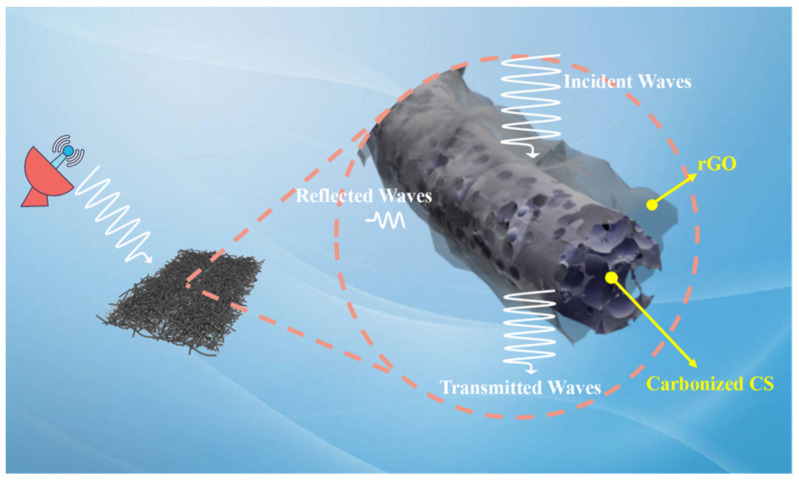
Carbon fiber wave absorbing schematic diagram.

**Table 1 nanomaterials-14-00587-t001:** XPS atomic ratios of PCNF-900.

Atom	Atomic%
C	91.36
N	2.67
O	5.95

## Data Availability

Data are contained within the article and Appendix A.

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
