# Peer review of "Reduced Graphene Oxide Modified Nitrogen-Doped Chitosan Carbon Fiber with Excellent Electromagnetic Wave Absorbing Performance"

_nanomaterials, 2024, doi:10.3390/nano14070587_

Round 1

Reviewer 1 Report

Comments and Suggestions for Authors

Decision: Minor revision

The manuscript is well-written and can be published after some minor corrections.

1. Line 97, please include a reference for the modified Hummer’s method.

2.      Line 99, ‘ultrasonography’ or ‘ultrasonification’?

3.      Line 121, specification for UV oven?

4.      Line 143, how many times bending and folding?

5.      Why topographical analysis and XPS were performed only for PCNF-900? How about the other two samples, i.e., PCNF-800 and PCNF-1000?

6.      Line 169, what kind of gas bodies?

7.      Lines 167-172, please explain why this is happening.

8.      Explain why rGO shedding increased with the increased carbonization temp.

9.   Be consistent with the sample nomenclature.

10. Please explain all samples with a 10% mass fraction of carbon fiber performed the best (lines 235-236; 250-251 and 269-271)?

11. Line 347, would it be better to call it a schematic diagram than a simulation diagram?

Comments on the Quality of English Language

English needs to be improved. Some sentences need re-framing throughout the manuscript.

Reviewer 2 Report

Comments and Suggestions for Authors

The subject of the assessed scientific article submitted to the journal Nanomaterials is related to the design of a new material with a dominant absorption coefficient of electromagnetic radiation. The presented article is interesting, but it leaves an impression of dissatisfaction in several respects:

1. In the experimental part (number 2), there is a Materials subsection (2.1), but the Methods subsection is missing. I understand that this function was probably intended to be fulfilled by Chapter 2.5 - Characterization, but it completely lacks information related to the methodology of electromagnetic research. There is also no standard according to which these tests were carried out and no block diagram of the measurement station.

2. On page 7 (line 239) it is stated that the measurement range of reflection coefficient tests is 4.32-14.48GHz, however, in the measurement charts (Fig. 7, 8, 9) the measurement range is 2-18GHz. Which range is correct? The question is also why the authors chose this and not another measurement range?

3. Analyzing the reflectance results in Fig. 7, it can be concluded that the relationship in graphs a, b, c is not proportional to the values of 5wt%, 30wt% and 10wt%. The proportional order should be the order of the charts: b (reflection loss ca 0dB), a (avarage reflection loss ca -8dB), c (avarage reflection loss ca -15dB) and should be appropriately correlated with the content of 5wt%, 10wt% and 30wt% or in the opposite direction: 30wt%, 10wt% and 5wt%. Was any error made during the analysis or processing of the results? The same remark also applies to the graphs in Fig. 8 and Fig. 9, where disproportion is also visibl.

4. A very important note related to the possibility of making a substantive error concerns the measurement of the reflection loss coefficient and the interpretation of these results. Based on the measurement of the reflection loss coefficient, the authors conclude on the value of the absorption coefficient of the tested material. This is probably an error, because the process of interaction of an electromagnetic wave with the tested material is related to three phenomena: transmission of an electromagnetic wave, reflection of an electromagnetic wave and absorption of an electromagnetic wave. When a wave falls on a material, part of it passes through the material, part is reflected and part is absorbed. The sum of these three values is 100%. If the authors only test the reflection loss and talk about the absorption coefficient content based on it, they completely neglect information about the transmission coefficient of the electromagnetic wave through the tested material !

5. Another issue is the frequency selectivity phenomena visible in the reflection coefficient waveforms (Fig. 7ac, 8ac, 9ac). The authors do not explain why different, shifted peaks appear on the graphs for different sample thickness values?

6. Why does chart c in Figures 7, 8, 9 always have a different sample thickness range than charts a and b? How can samples with different parameters (e.g. thickness) be compared with each other?

Reviewer 3 Report

Comments and Suggestions for Authors
The authors presented carbon fibre based rGO/CF composites in chitosan for electromagnetic wave absorber application. While the data are clear to understand, the research lacks few critical points. The references were not chosen carefully and I recommend to include several high impact publications rather than average output data.  Please include the following papers which are highly relevant (ACS Appl. Nano Mater. 2023, 6, 15, 14245–14254; ACS applied materials & interfaces 9 (3), 2017, 3030-3039;Physica B: Condensed Matter 2014, 448, 73-76.)

The authors described their composites as an absorber, however, reflection loss alone cant determine the absorption coefficient of EM wave. It is clear from the terminology itself. to determine the EM absorption, we need total shielding effectiveness value. 

The explanation of results were not found. The conc. of 10% filler system produced maximum reflection loss in fig. 7. But no explanation provided. Same for others. What happened with that conc. whereas higher weight% of filler actually led to very low attenuation. In my opinion, CNT/rGO system cant significantly contribute to absorption mechanism. I recommend the SEtotal values and from there, authors have to determine effective absorption coeff. 

Finally, EM wave absorption is not the case here, even if so, its not the dominant mechanism. Magnetic loss cant effectively determine absorption. Dielectric loss from the filler alone cant provide absorption. It is mostly reflection-based process here.
Comments on the Quality of English Language

Moderate checks required. Several English issues found in introduction section. 

Reviewer 4 Report

Comments and Suggestions for Authors

The authors need to add the answers to the following questions to the manuscript:

11.       How did the authors control parameters to ensure uniformity during electrostatic spinning, electrostatic self-assembly, and high-temperature carbonization?

22.       How did the authors optimize carbonization temperature, graphene coating thickness, N-doping level, and filler content to achieve the desired electromagnetic wave absorption properties?

33.       I would suggest that the authors discuss more about the polarization, conductivity, and magnetic properties at the nanoscale level and correlate them to the material structure.

44.       How is the current work different from similar work with rGO such as:

https://doi.org/10.1557/s43577-023-00538-z? Provide details.

55.       How can the authors ensure the scalability and reproducibility of their synthesis process is for practical applications?

66.       It would be nice if the authors could discuss the environmental impact of the synthesis process, the cost of materials and production, and the availability of raw materials for the practical viability of their proposed assembly in real-world applications.

77.       While the study demonstrates promising electromagnetic wave absorption properties in a laboratory setting, there is no information provided for the durability, compatibility, and overall performance once it is integrated into practical applications such as radar-absorbing coatings or electromagnetic interference shielding. Please discuss about this.

Comments on the Quality of English Language

As above.

Round 2

Reviewer 2 Report

Comments and Suggestions for Authors

My comments and remarks on the authors' corrections and replies:

Ad1. The attached information about the measurement method, located in supporting information, generates more questions than answers:

- Fig.2 (why number two and not one) in the supporting information does not show the measurement station at work and the way of placing the tested sample.

- if measurements are performed in the 2-18GHz range, why is a coaxial ring used and not the waveguide measurement method? In this measurement range, we are already dealing with an electromagnetic wave (which is excited in the TE or TM modes) and only waveguide measurement or measurement in open space or using a measurement window can provide good results.

- why, when using such an advanced device as VNA, was the reflectance calculated based on formulas and not simply measured? After all, this technologically advanced measuring equipment probably allows you to measure the transmission coefficient and reflection coefficient with direct measurement of the module and the phase of these quantities.

- why was "theoretical simulation of the transmission line" used with such an advanced measuring device as VNA Keysight? Do the authors really know what measurements they perform and know how to use measuring equipment?

- the description in supporting information contains many errors and inaccuracies and I suggest that the authors make serious measurement errors.

Ad2. I understand that the measurements were performed in the range of 2-18GHz?

Ad3. The response to comment 3 proposed by the authors does not actually explain anything. If it is actually the case that the RL results are ordered a, b, c as in the new figures 7, 8, 9, the question is why is this happening? The non-linearity of the RL coefficient depending on the wt% of the sample is visible here. What about intermediate %wt values or smaller or larger values than the current %wt values? These three values of 5, 10, 30%wt are definitely too little to determine the nature of RL changes. Maybe for the value of 20%wt it will be even better?

Ad4. The answer to one of the most important questions – question no. 4 – is unsatisfactory. Unfortunately, the information contained in the supporting information shows that the authors perform very strange measurements using an advanced VNA device. What exactly do the authors mean when they use the statement "proprietary mode"?

Ad5. From my knowledge, the shifting of peaks in the RL characteristic should be related to its geometry, not thickness. Are the samples homogeneous? The thickness of a homogeneous sample can only affect the height of the peak itself and not change its frequency

Ad6. Why was the title of the article changed to "...with Excellent Electromagnetic..."? Where can you see these wonderful properties?

Reviewer 3 Report

Comments and Suggestions for Authors

Can be accepted now

Author Response

We thank the reviewer for the positive feedback.

Reviewer 4 Report

Comments and Suggestions for Authors

N/A

Comments on the Quality of English Language

As above.

Author Response

Thank the reviewer for their time and effort in reviewing themanuscript.

Round 3

Reviewer 2 Report

Comments and Suggestions for Authors Although I am still not convinced about the measurement issues and the obtained
measurement results and their results, I can agree with the opinions of the rest
of the reviewers who accepted the article for publication. I can also suggest that the editor seek the opinion of another reviewer who will
make the final decision.